# Identification and Expression Profiling of the *Regulator of Chromosome Condensation 1 (RCC1)* Gene Family in *Gossypium Hirsutum* L. under Abiotic Stress and Hormone Treatments

**DOI:** 10.3390/ijms20071727

**Published:** 2019-04-08

**Authors:** Xiao Liu, Xingchen Wu, Chendong Sun, Junkang Rong

**Affiliations:** 1The State Key Laboratory of Subtropical Silviculture, College of Forest and Biotechnology, Zhejiang Agricultural and Forestry University, Hangzhou 311300, China; 15068170595@163.com; 2The Key Laboratory for Quality Improvement of Agricultural Products of Zhejiang Province, College of Agriculture and Food Science, Zhejiang Agricultural and Forestry University, Hangzhou 311300, China; singchen@aliyun.com

**Keywords:** cotton, *RCC1*, gene family, gene expression, salt tolerance

## Abstract

The regulator of chromosome condensation 1 (RCC1) is the nucleotide exchange factor for a GTPase called the Ras-related nuclear protein, and it is important for nucleo-plasmic transport, mitosis, nuclear membrane assembly, and control of chromatin agglutination during the S phase of mitosis in animals. In plants, RCC1 molecules act mainly as regulating factors for a series of downstream genes during biological processes such as the ultraviolet-B radiation (UV-B) response and cold tolerance. In this study, 56 genes were identified in upland cotton by searching the associated reference genomes. The genes were found to be unevenly distributed on 26 chromosomes, except A06, A12, D03, and D12. Phylogenetic analysis by maximum-likelihood revealed that the genes were divided into five subgroups. The *RCC1* genes within the same group shared similar exon/intron patterns and conserved motifs in their encoded proteins. Most genes of the *RCC1* family are expressed differently under various hormone treatments and are negatively controlled by salt stress. *Gh_A05G3028* and *Gh_D10G2310*, which encode two proteins located in the nucleus, were strongly induced under salt treatment, while mutants of their homoeologous gene (*UVR8*) in *Arabidopsis* and VIGS (virus induced gene silencing) lines of the two genes above in *G. hirsutum* exhibited a salt-sensitive phenotype indicating their potential role in salt resistance in cotton. These results provide valuable reference data for further study of *RCC1* genes in cotton.

## 1. Introduction

Cotton (*Gossypium* spp.) is an important economic crop worldwide. The demand for cotton production in China is high and the arable land area available to these crops decreases each year. Due to insufficient farm land for food crops, cotton is most often planted on saline-alkali and dry land. Abiotic stress, especially high salinity, restricts the productivity and quality of cotton. Although cotton is moderately salt-tolerant, seedlings are severely inhibited under high salt stress and the yield is seriously affected [1]. Like salt stress, drought inhibits cotton growth and development, leading to a decrease in yield [2,3].

High concentrations of K^+^ and Na^+^ in the growth environment impact cellular osmosis, ionic homeostasis, and photosynthesis, thus resulting in the generation of reactive oxygen species (ROS), damage to membrane systems, and ultimately, apoptosis [4,5,6]. Cellular ROS, derived under saline stress, have adverse effects on plant cells and cause lipid peroxidation, DNA damage, protein denaturation, carbohydrate oxidation, pigment breakdown, and impairment of enzymatic activity [6]. Elimination and neutralization of ROS relies on cellular antioxidants and accumulation of three enzymes, including superoxide dismutase, peroxidase, and catalase [6]. Moreover, unsaturated fatty acids in membrane lipids protect the photosynthetic machinery against damage under salt-induced osmotic stress [7].

Plants harbor a series of mechanisms to cope with adverse effects under salt stress, such as osmotic adjustment, osmo-protection, sodium exclusion, sodium sequestration, and potassium retention [4,6]. Soluble sugars, such as sucrose and glucose, play a crucial role in cellular respiration and maintenance of osmotic potential, providing the ability to imbibe moisture under salt stress [7].

The *regulator of the chromosome condensation 1 (RCC1)* gene family is important to the cell cycle. Although these genes have been studied extensively in mammals, no studies in cotton have yet been reported. The RCC1 is the nucleotide exchange factor for a GTPase called the Ras-related nuclear protein (Ran). The RCC1 was first discovered in 1987 and is considered to be a regulator of chromosome concentration. It encodes the 45 kDa nuclear protein whose sequence is highly conserved among all eukaryotes and consists of seven beta helices, also known as seven RCC1 repeat units [8]. In the nucleus, RanGDP is converted to RanGTP under the catalysis of RCC1, which plays an important role in the assembly of spindles during mitosis, prevention of multiple replication of S-phase DNA, transport of nuclear material, and nuclear membrane reconstruction [9,10,11]. Due to its ability to prevent multiple replication of S-phase DNA, many studies in humans have shown that *RCC1* is associated with cancer. For example, *RCC1* expression is inhibited in gastric cancer cells [12].

In plants, the first RCC1 family protein to be identified was UVR8 (UVB-resistance protein 8) in *Arabidopsis* [13]. UVB-resistance protein 8, the only UV-B receptor in plants, is responsible for regulating expression of a series of self-protection genes under UV-B and is involved in photo-morphism. To date, the functional study of UVR8 in *Arabidopsis* has made some progress. Normally, UVR8 is evenly distributed in the cytoplasm and nucleus; however, under UV-B treatment, it tends to accumulate in the nucleus through interaction with constitutive photomorphogenic 1 (COP1), thus triggering a UV-B cascade [14,15,16,17,18].

The mitochondrial protein RUG3 (RCC1/UVR8/GEF-like3) in *Arabidopsis* also belongs to the RCC1 protein family and is involved in the splicing of nad2 messenger RNA in mitochondria and the formation of nicotinamide adenine dinucleotide (NADH) complexes. RCC1/UVR8/GEF-like3 is a member of the RCC1 family of seven basic structural units. Unlike UVR8, RUG3 localizes in mitochondria and has no homologous genes in aquatic plants, animals, or fungi. The RUG3 is a terrestrial plant-specific RCC1 structural protein [19]. Tolerant to chilling and freezing 1 (TCF1) encodes an RCC1 family of proteins consisting of six RCC1 basic repeat units, which are highly similar to UVR8 and human RCC1 protein structures. Tolerant to chilling and freezing 1 plays an important role in regulating plant tolerance to low temperatures [20]. The *PRAF* (PH (pleckstrin homology domain), RCC1 and FYVE ((Fab 1, YOTB, Vac 1 and EEA1) zinc-finger domain)) gene family is a subfamily of the plant *RCC1* gene family. It is also a unique gene family in plants containing special domains and specific sequences [21]. The MtZR1 protein is a typical PRAF protein in sputum and is expressed in root meristem and vascular bundle tissue of mature stem segments [22]. The expression of its gene in roots is negatively regulated by a variety of abiotic stresses [7].

To fully understand the possible function and evolution of the *RCC1* gene family in four species of cotton, we identified 174 *RCC1* genes from the cotton genome [23], including 30 from *G. arboretum*; 29 from *G. raimondi*; 59 from *G. barbadense*; and 56 from *G. hirsutum*. Phylogenetic analysis was carried out for the four species and the genetic structure and conserved motifs of *G. hirsutum* genes were studied. Tissue specificity was analyzed by real-time quantitative reverse transcription (qRT) PCR and the expression of 56 *GhRCC1* genes under abiotic and hormone stress was investigated.

## 2. Results

### 2.1. Comparison of RCC1 Genes Among Four Cotton Species

To identify *RCC1* genes in *G. arboreum*, *G. raimondii*, *G. barbadense*, and *G. hirsutum*, HMMER and BLAST searches were performed using 25 *RCC1* genes from *Arabidopsis* as the query. A total of 174 *RCC1* gene sequences were identified, of which 30 were *G. arboreum* sequences, 29 were *G. raimondii* sequences, 59 were *G. barbadense* sequences, and 56 were *G. hirsutum* sequences, and the number of *G. hirsutum* is almost the accumulation of *G. arboreum* and *G. raimondii*, and the same is true for *G. barbadense* (Figure 1A). We also obtained the number of RCC1 family genes in cocoa and grape, both of which contain 21 genes. The results indicated that the RCC1 gene family showed no preference between the varieties in evolution and the RCC1 family gene is highly conserved. To get a better understanding of phylogenetic relationships and biological functions of RCC1 genes, the amino acid sequences of the RCC1 family genes of *G. arboreum*, *G. raimondii*, *G. barbadense*, and *G. hirsutum* were extracted from the published genome database.

### 2.2. Phylogenetic Analysis of the RCC1 Gene Family

Phylogenetic tree analysis by Clustal Omega and constructed by iTOL v4 revealed that the RCC1 family genes are divided into five subgroups (Group I, II, III, IV, and V). Each subgroup includes the RCC1 family genes of *G. arboreum*, *G. raimondii*, *G. barbadense*, and *G. hirsutum*. Interestingly, the number of RCC1 genes in each subgroup from every species was approximately equal. For example, Group III contained one gene each from GaRCC1 and GrRCC1 and two each from GbRCC1 and GhRCC1. The distribution of genes among the groups was not uniform. The lowest number of genes were in Group III (6 genes) and the highest number were in Group V (69 genes) (Figure 1B).

### 2.3. Exon–Intron Structure, Motif Assay, and Chromosomal Location Analyses

The analysis of the gene structure including intron–exon size and number indicated that different members of the RCC1 family (Figure 2) contain varied exons from 3 to 16. It is worth noting that the members with close relatives are similar in exon–intron structure and the most important differences among them are the lengths of the exon–intron. *Gh_D09G1044, Gh_A09G1023*, and *Gh_D08G2751* all have nine exons, and the fragment length of the sixth exon is the longest. *Gh_A13G1460* has only three exons. *Gh_A07G0164, Gh_D07G0221, Gh_D11G1503*, and *Gh_A11G1355* all contain four exons, but exon length is clearly divided into two groups.

To further investigate the conservation of motifs in RCC1 in upland cotton, protein sequences of 56 GhRCC1 genes were identified and 20 conserved motifs were identified (Figure 3). Closely related genes have similar conserved motifs (*Gh_A08G0100*, *Gh_D08G2751*, *Gh_A09G1023*, *Gh_D09G1044*, *Gh_D10G1907*, *Gh_A10G1652, Gh_A13G0645*, *Gh_D13G0764*, *Gh_A03G0868*, *Gh_D02G1249*, *Gh_D11G0331*, and *Gh_A11G0277*). These genes all contain the same motif (3, 4, 6, 9, 10, 12, 13, 15), whereas *Gh_D09G0059*, *Gh_A09G0062*, *Gh_D10G1050*, and *Gh_A10G0603* contain the same motif (2, 3, 5, 17), which proves that the RCC1 gene family sequence is highly conserved. Although the motifs of closely related genes are similar, the size of the gene fragments varies widely (947–18,537 bp), such that the *Gh_A08G0100* gene fragment is much smaller than *Gh_D08G2751*.

Based on the complete genome sequence of upland cotton, we used the MG2C website to map all 56 genes to chromosomes (Figure 4). Among them, 26 genes were located on chromosome A and 28 on chromosome D. Of the 56 genes, *Gh_D08G2751* and *Gh_A11G3194* were not mapped to chromosomes but to scaffolds. The largest number of genes were located on chromosome D11 (seven genes), followed by chromosome A11 (six genes), chromosome D05 (five genes), and chromosome A09 (four genes), while A06, A12, D03, and D12 contained none of the genes. Other chromosomes contained between one and three of the genes.

### 2.4. Analyses of Tissue-Specific Expressions

To explore the biological function of *GhRCC1* genes, qRT-PCR was used to determine the spatial specificity expression pattern of 56 *GhRCC1* genes in fourteen cotton organs. As indicated in Figure 5, some *GhRCC1* genes were differently expressed in the fourteen tissues tested, while other *GhRCC1* genes showed similar expression patterns in diverse tissues, which may manifest from functional differentiation of *GhRCC1* genes during plant development. For example, *Gh_D01G0233*, *Gh_D13G0764*, and *Gh_D07G0221* were preferentially expressed in terminal buds, implying that these genes may play regulatory roles at flower bud differentiation stages, while *Gh_D13G1736*, *Gh_A04G0246*, *Gh_A11G1355*, and *Gh_D08G0551* were highly expressed in 0 DPA tissue, so these genes may be involved in fiber initiation. In general, the *GhRCC1* genes that are highly expressed in specific tissues may be involved in the regulation of plant development. *Gh_D11G2263*, *Gh_A11G0277*, and *Gh_A11G2084*, for instance, were highly expressed in the ovule 30 DPA, suggesting that they might be related to the elongation of the developing fibers in the seed coat. Conversely, some genes are not expressed specifically. Furthermore, *Gh_A05G3028* and *Gh_A10G0603* were relatively highly expressed at each stage, suggesting that they may be involved in the development of every tissue.

### 2.5. Expression of GhRCC1 Genes in Response to Various Hormone Treatments

Phytohormones play a critical role in various biological processes throughout all the stages of vegetative growth and reproductive growth. Research focusing on the development of cotton fiber has revealed the important roles of auxin, gibberellic acid, and ethylene in fiber initiation and elongation [24,25,26,27,28,29,30,31].

Aminocyclopropane-1-carboxylic acid (ACC), a precursor of ethylene, was used for ethylene due to its relative stability. To identify the *GhRCC1* genes responsive to auxin, gibberellic acid, and ethylene, we analyzed the expression profiles under IAA (indole-3-acetic acid), GA (gibberellic acid) and ACC (1-aminocyclopropane-1-carboxylic acid) treatment (Figure 6). Under these phytohormone treatments, most of the RCC1 genes showed different expression patterns, and even the expression of the same gene in the aerial and underground parts was very different. In the aerial parts treated for 3 h with hormones, most of the genes were downregulated or unaffected. Gene expression was only slightly upregulated in the ethylene or GA pathways; for example, under ethylene treatment, *Gh_D02G0718* and *Gh_D04G0616* expression increased, while under GA treatment, *Gh_A11G1503* and *Gh_D11G1664* increased. Interestingly, the expression of these four genes was also elevated in the underground parts of the plant. In fact, in the underground parts, most of the genes, including *Gh_A03G1146, Gh_A10G0603,* and *Gh_A09G0062*, were upregulated under treatment with ethylene and GA. The relevant expression data were downloaded from the ccNET website. *Gh_D02G0718, Gh_A11G1503,* and *Gh_D11G1664* were highly expressed in the ovules one day before flowering, while *Gh_A10G0603* and *Gh_A09G0062* were specifically expressed in 1 DPA and 35 DPA. These results indicate that *Gh_D02G0718, Gh_A11G1503, Gh_D11G1664*, *Gh_A10G0603,* and *Gh_A09G0062* are probably involved in GA and ethylene pathways to control fiber development.

### 2.6. Expression Patterns of GhRCC1 Genes in Response to Various Abiotic Stress Treatments and Subcellular Location

To probe their potential roles in the cotton response to various abiotic stresses, expression levels of GhRCC1 genes under 200 mM NaCl and 17% PEG6000 stress treatments were determined by qRT-PCR (Figure 7). Under the NaCl treatment, most RCC1 genes showed a negative regulatory pattern and only 14% of the genes were upregulated in shoots. Following the PEG6000 treatment, the expression levels of *Gh_A05G3028*, *Gh_D10G2310*, *Gh_A10G2003*, *Gh_D04G0616*, *Gh_D13G1736*, *Gh_A11G2084*, *Gh_A07G0164*, *Gh_D11G2399*, *Gh_A03G1146*, *Gh_A10G0603*, *Gh_D05G2447*, *Gh_A02G0670*, and *Gh_D02G0718* were increased in both shoots and roots; *Gh_A09G2009*, *Gh_D09G2222*, *Gh_A13G0645*, and *Gh_D13G0764* were downregulated in both shoots and roots; and *Gh_D11G2263*, *Gh_A09G0062*, and *Gh_D10G1050* were upregulated in shoots and downregulated in roots.

As the results showed that *Gh_A05G3028* and *Gh_D10G2310* were dramatically upregulated under salt treatment, mutants of their homologous genes in *Arabidopsis* were used to determine their role in the control of salt resistance of cotton. Although the amino acid sequences of Gh_A05G3028 and Gh_D10G2310 are best matched to TCF1 in *Arabidopsis*, *TCF1* was minimally expressed unless under low temperatures. Thus, T-DNA mutants of another homologous gene, *UVR8*, were identified (Appendix A) and used in phenotypic analyses of salt treatment. Our results showed that mutants of UVR8, *uvr8-1* and *uvr8-2*, exhibited salt-sensitive phenotypes in comparison with the wild-type (Figure 8A) and the chlorophyll content in the leaves of *UVR8* mutants under salt treatment decreased more significantly comparing with the wild type of *Arabidopsis* (Figure 8B), further indicating a potential role for *Gh_A05G3028* and *Gh_D10G2310* in the salt resistance of cotton. To conform their presumptive biological function of *Gh_A05G3028* and *Gh_D10G2310* in salt resistance, these two genes were silenced using VIGS (virus induced gene silencing) system. Seven days after infection, *TRV2: GhCLA* cotton exhibited an albino phenotype (Figure 8C), indicating that the VIGS system was already working. Meanwhile, the expression level of *Gh_A05G3028* and *Gh_D10G2310* in negative control (TRV1 + TRV2) and their VIGS lines (TRV1 + TRV2: Gh_A05G3028 and TRV1 + TRV2: Gh_A05G3028) were detected by RT-qPCR. As the results showed, the expression level of the two genes in VIGS interfering lines was sharply decreasing in comparison to the negative control (Figure 8D). After a 12-day salt treatment, the *Gh_A05G3028* and *Gh_D10G2310* VIGS interfering lines appeared to wilting while the negative control showed a resistance to salt stress (Figure 8E). All the results above proved the crucial roles of *Gh_A05G3028* and *Gh_D10G2310* in the salt resistance of cotton.

In *Arabidopsis*, RCC1 family proteins located in the nucleus, such as UVR8 and TCF1, as homologous proteins of Gh_A05G3028 and Gh_D10G2310, play a crucial role in activating a series of downstream genes under UV-B or cold stress. The results of subcellular localization assays in epidermal cells of tobacco of Gh_A05G3028 and Gh_D10G2310 confirmed that these proteins are distributed in the nucleus (Figure 9), which implies that *Gh_A05G3028* and *Gh_D10G2310* control the expression of several genes under salt stress. The ccNET was used for co-expression analysis of the two genes and this identified a series of genes which were positively or negatively co-expressed with *Gh_A05G3028* and *Gh_D10G2310*. The GO and KEGG analyses of these co-expression genes were used to determine genes related to salt resistance. Among them, genes such as *Gh_A05G3863, Gh_A10G2136, Gh_D05G1179*, and *Gh_D10G2457* have oxidoreductase activity, which acts on the paired donor. The oxidation of a pair of donors causes oxygen molecules to be reduced to two molecules of water. The three genes *Gh_A08G2526, Gh_D08G2762*, and *Gh_D08G2764* are related to chlorophyll coenzyme A; *Gh_A02G1698, Gh_A13G1883*, and *Gh_D03G0021* are closely related to the photorespiration process; *Gh_A05G3863, Gh_A10G2136, Gh_D05G1179, Gh_D10G2457, Gh_A13G1619, Gh_D06G0402,* and *Gh_D13G1979* are related to the biosynthesis and metabolism of unsaturated fatty acids; while *Gh_A05G2107, Gh_D05G2362,* and *Gh_D10G1386* are involved in the sugar metabolism pathway. Therefore, we speculate that *Gh_D10G2310* and *Gh_A05G3028* respond to salt stress mainly by affecting oxidoreductase activity, cell membrane stability, and cell osmotic pressure. The expression data of the above 16 genes involved in reactive oxygen species signaling, sugar metabolism, saturated fatty acid metabolism, and unsaturated fatty acid synthesis under salt stress was downloaded from ccNET (Appendix A). The FPKM (fragments per kilobase of exon per million reads mapped) from the database of ccNET revealed that 10 genes among them were upregulated under salt stress in comparison to the control (Appendix A). Nevertheless, 1 gene among the 10 genes is hardly expressed under control and salt treatment. (FPKM < 10) (Appendix A). To further confirm the expression profile of these nine genes and the two candidate RCC1 family genes above under salt stress, we conducted RT-qPCR. The results showed that the above related genes were upregulated under salt stress (Figure 10), which proved that the transcriptome data were basically true and reliable. Based on the above analysis, we believe that Gh_D10G2310 and Gh_A05G3028 proteins may affect REDOX activity, cell membrane stability, and osmotic pressure by participating in the reactive oxygen species signal pathway, glucose metabolism, and other signal transduction pathways, and thus play a positive regulatory role in the cotton salt stress response.

## 3. Discussion

As a grain and oil crop, cotton plays an important role in industrial and agricultural production worldwide [21,32]. In animals, RCC1 is responsible for activating Ran and linked to many biological processes coupled with Ran, such as nucleo-cytoplasmic transport, mitosis, nuclear-envelope assembly, repair of DNA damage, and incidences of cancer [8,9,11,33,34,35,36,37,38]. The RCC1 family members in plants can be grouped into two major categories: one consisting of 6–7 RCC1 repeat units, which is similar to human RCC1, the other composed of multi-domains, including the RCC1 repeats domain [19]. However, the biological role of RCC1 family genes in plants remains unclear. According to assessments of UVR8 and TCF1 in *Arabidopsis*, monodomain RCC1 family genes tend to be involved in regulating signal cascades, for example, the UV-B and cold-induced signal pathways [20,39,40,41,42]. To characterize the RCC1 family of genes in the heterotetraploid species *G. hirsutum*, we performed a comprehensive analysis of GhRCC1 genes, including studies of phylogenetic relationships, gene structure, conserved motifs, chromosomal location, and expression profiles in different tissues. In recent years, the genome-wide sequencing of the four cotton species has been completed, which has greatly contributed to the development of cotton science. In this context, 174 RCC1 family genes were detected, including 56 GhRCC1 genes, 59 GbRCC1 genes, 29 GrRCC1 genes, and 30 GaRCC1 genes. Based on phylogenetic relationships with *G. arboreum*, *G. raimondii*, *G. barbadense*, and *G. hirsutum*, the RCC1 family was divided into five subfamilies. In addition, we found that the genomes of *G. barbadense* and *G. hirsutum* are twice as large as those of *G. arboreum* and *G. raimondii* and have the same pattern in five subgroups, which is consistent with the evolution of heterotetraploids. Most RCC1 genes have >8 exons and only a few have <5 exons, indicating that RCC1 family genes may be regulated by two different mechanisms.

There is increasing evidence that introns play an important role in the production of non-coding RNA and selective splicing [17]. According to the specific expression of RCC1 family genes in tissues, *Gh_D01G0233, Gh_D13G0764*, and *Gh_D07G0221* were found to be preferentially expressed in the terminal bud, and *Gh_A05G3028* and *Gh_A10G0603* genes were more highly expressed than other genes in each organ. The results indicate that RCC1 family genes play a crucial role in the growth and development of cotton. At the same time, we used the *Arabidopsis Gh_D10G1050* homologue *UVR8* to conduct salt stress experiments and found that it is salt-sensitive. However, the specific regulatory relationship between the RCC1 family genes and salt stress is still unclear and further functional exploration is needed. As previously reported, plant hormones, such as auxin, GA, and ethylene, are important to the growth and development of cotton. This study showed that the expression of most genes increased in the underground part after GA and ethylene treatment, and thus it is speculated that they have a potential role in the development of cotton fibers.

Under unfavorable conditions such as extreme temperatures, drought, and high salt, a series of physiological metabolic reactions occur in the plant body, which manifest as reversible inhibition of metabolism and growth or irreversible damage when serious, leading to the death of the whole plant. Among all kinds of stresses, drought and salt damage have the most serious impact on potential crop yields [43]. Compared with other major crops, cotton has a relatively high salt tolerance; nevertheless, in highly saline soil, the growth, yield, and quality of cotton is seriously affected, especially during the germination and seedling stages. Stress induces gene expression, which leads to the accumulation of certain substances and changes in metabolic pathways, and in turn allows plants to adapt accordingly [44]. Generation of cell intracellular ROS, as well as damage to cell membranes and DNA integrity, resulting from salinity can lead to apoptosis [4,6]. Thus, synthesis of soluble sugars, peroxisomes, and unsaturated fatty acids become three critical mechanisms in maintaining the ability to absorb water, retain membrane integrity, and eliminate ROS [4]. To determine the role of RCC1 family genes in salt resistance, we detected the expression profile of RCC1 family genes in *TM-1* under salt stress by RT-PCR. *Gh_A05G3028* and *Gh_D10G2310*, homologous genes of *TCF1* and *UVR8*, were sharply induced under salt treatment. Additionally, we conducted a salt treatment assay using the T-DNA insertion mutants of *UVR8* in *Arabidopsis* and VIGS lines of *Gh_A05G3028* and *Gh_D10G2310* in *G.hirsutum*. As the results, both *UVR8* mutants and VIGS lines of *Gh_A05G3028* and *Gh_D10G2310* exhibited a salt-sensitive phenotype, further confirming the potential role of *Gh_A05G3028* and *Gh_D10G2310* in control of salt resistance. As is known, TCF1 and UVR8 trigger a series of genes in the UV-B and cold signal pathways that alter plant resistance in these conditions [20,39,40,42]. In addition, positive co-expression of *Gh_A05G3028* and *Gh_D10G2310* was determined by KEGG pathway enrichment analysis. Genes involved in biosynthesis of unsaturated fatty acids, sucrose synthesis, metabolic processes, and encoding peroxisomes were identified and analyzed to reveal a close relationship with *Gh_A05G3028* and *Gh_D10G2310*. Hence, *Gh_A05G3028* and *Gh_D10G2310* probably adjust the resistance of cotton to salt stress though epigenetic regulation of the salt resistance-related genes above, as the homologous gene, *TCF1* in *Arabidopsis*, reduces target gene expression by altering levels of H3K4me3 and H3K27me3 [20].

## 4. Materials and Methods

### 4.1. Plant Materials and Treatment

Seeds of *Arabidopsis thaliana (L.) Heynh Col-0* (wild type), *uvr8-1* (SALK_072594C), and *uvr8-2* (SALK_072594C) from the Nottingham Arabidopsis Stock Centre (NASC) (http://arabidopsis.info/BasicForm) were germinated and grown on 1/2 MS medium (pH 5.8) for 7 days under the following conditions: 12,000 Lx light for 16 h at 25 °C/dark 8 h at 23 °C, 80% humidity, and then transferred into 1/2 MS medium and 1/2 MS medium containing 150 μm NaCl, pH 5.8 under the same condition for 3 days. Leaves of *Arabidopsis thaliana* above were collected and used for determination of chlorophyll content.

For phytohormone treatment, seeds of *TM-1* (Texas Marker-1, the upland cotton genetic standard line) were germinated and cultivated under the conditions above in wet vermiculite, and then soaked in 1/2 MS liquid medium containing 50 μm indole-3-acetic acid (IAA), 5 μm gibberellic acid (GA), and 50 μm aminocyclopropane-1-carboxylic acid (ACC) for 3 h, respectively. For salt and PEG6000 treatment, ten-day-old seedlings of *TM-1* were soaked in 1/2 MS liquid medium with 200 μm NaCl for 3, 6, and 12 h; and 17% PEG6000 for 3 h. The *TM-1* soaked in normal 1/2 Murashige and Skoog (MS) liquid medium was used as the control. Samples of 7-, 14-, and 30-day-old seedling shoots and roots, 60-day-old leaves, stems, and buds, 0, 10, and 30 DPA (day post-anthesis) ovules, and 30 DPA fibers were harvested from the *TM-1* growth in soil at an average temperature of 30 °C.

To further confirm the expression pattern of *Gh_A05G3028* and *Gh_D10G231* and their potential downstream genes under salt treatment, seeds of *TM-1* were germinated in soil under the conditions: 12,000 Lx light 12 h at 23 °C/dark 12 h at 23 °C, 80% humidity for 14 days. Then part of the plants were watered with 1/2 MS nutrient solution as the control and the others were watered with 200 mM salt solution regularly every 3 days until the phenotypes appeared. Cotton leaves of *TM-1* under control and salt treatment were collected for fluorescence quantitative PCR at 0, 1, and 3 h after treatment, respectively.

### 4.2. Identification of RCC1 Genes in Gossypium spp.

To identify members of the RCC1 gene family in *Gossypium spp*., *Arabidopsis* RCC1 sequences were obtained from the TAIR database (http://www.arabidopsis.org) [45] and used for a BLASTP algorithm-based query against the *Gossypium spp.* genome database (https://www.cottongen.org) [23]. Genes were identified by a hidden Markov model search based on the RCC1 domain using the Pfam protein domain database. Details of all *GhRCC1* gene are listed in Appendix A.

### 4.3. Phylogenetic Analysis

The amino acid sequences of the cotton and *Arabidopsis* RCC1 genes were extracted using Clustal Omega (https://www.ebi.ac.uk/Tools/msa/clustalo/) [46] for multiple sequence alignment. To construct an RCC1 protein phylogenetic tree using iTOL v4 (http://itol.embl.de/) [47], RCC1 proteins from *Arabidopsis*, *G. raimondii, G. arboreum, G. barbadense*, and *G. hirsutum* were used. The neighbor-joining method with amino acid p-distance was applied to construct the tree and reliability was obtained by bootstrapping with 1000 replicates.

### 4.4. Exon/Intron Structure, Motif Assay and Chromosomal Location Analyses

To better understand the evolutionary relationships between different members of the RCC1 gene family, the genomic DNA and cDNA sequences of predicted genes were obtained from the four cotton genomes. Multiple sequence alignment of all identified RCC1 proteins was performed using Clustal Omega with default parameters (https://www.ebi.ac.uk/Tools/msa/clustalo/).

The Gene Structure Display Server (http://gsds.cbi.pku.edu.cn/) mapped the exon–intron structure diagrams of individual genes in the *GhRCC1* family based on the full-length coding sequences and genomes in the cotton genomics database. The exon–intron structure maps of all upland cotton RCC1 genes were drawn using TBtools software (Dr Chengjie Chen, Guangzhou, China) based on the published Upland Cotton Genome Document and the ID of the *GhRCC1* gene family.

Conservative motifs in cotton *RCC1* were identified by the MEME suite program (http://meme-suite.org/tools/meme). The parameters were set as follows: the maximum number of motifs = 20; the minimum width of the pattern = 8; the maximum width of the pattern = 50. Other conditions were set to default values.

The chromosomal location information of all *GhRCC1* genes was derived from the annotation files downloaded from the CottonGen website. The MG2C website was used to distribute the GhRCC1 genes along the chromosome from the top to the bottom (http://mg2c.iask.in/mg2c_v2.0/).

### 4.5. RNA Isolation and Quantitative Real-Time Polymerase Chain Reaction (qRT-PCR)

Total RNA of plant materials was extracted from 14 different spatio-temporal organs using the RNAprep Pure Plant Kit (Polysaccharides & Polyphenolics-rich) (TIANGEN, Beijing, China) according to the manufacturer’s instructions.

The RNA quantity and purity were assessed using a NanoDrop One spectrophotometer (NanoDrop Technologies, Wilmington, DE, USA). Total RNA was reverse-transcribed by Reverse transcriptase M-MLV (Takara, Beijing, China). The reaction system and procedures were as follows: Oligo dT 8 μL, dNTPs 4 μL, RNase-Free ddH_2_O 38 μL, total RNA 8 μL, after mixing, put in a PCR instrument at 68 °C for 10 min, ice bath for 10 min, then add RRI (RNase Inhibitor) 2 μL, 5 × MLV buffer 16 μL, M-MLV 4 μL to the centrifuge tube, mix and mix, and put into the PCR instrument (42 °C, 60 min; 70 °C, 15 min; 4 °C, ∞). The qRT-PCR was then performed using a 7300 Real-Time PCR System (Applied Biosystems, Foster City, CA, USA) according to the supplier’s protocols. Each reaction mixture contained 6 μL DNase/RNase-free water, 11 μL TB Green Real-Time PCR master mix, 2 μL diluted cDNA product from the reverse transcription PCR reaction, and 1 μL gene-specific primer. Three biological replicates were conducted for each tissue and each biological replicate was technically repeated three times. All primers are listed in Appendix A. The thermal cycle was as follows: 95 °C for 30 s followed by 45 cycles of denaturing at 95 °C for 15 s and annealing and elongating at 58 °C for 30 s. The expression values of the RCC1 genes were normalized with an internal reference gene *UBQ*. The relative expression levels were calculated using the 2−∆∆Ct method. A heat map for gene expression patterns was generated using OmicShare tools, a free online platform for data analysis (http://www.omicshare.com/tools).

### 4.6. Identification of T-DNA Insertional Mutants

The DNA of *Arabidopsis* rosette leaves was extracted using the TPS method. Primers for identification of *uvr8-1* (SALK_072594C) and *uvr8-2* (SALK_072594C) were designed from the website (http://signal.salk.edu/tdnaprimers.2.html) and are listed in Appendix A. Homozygous mutants were identified using the method as described by Ronan [48].

### 4.7. Subcellular Localization

Full length open reading frames of *Gh_A05G3028* and *Gh_D10G2310*, including upstream and downstream adaptors (21 bp matching to flanking sequences of multiple cloning sites in an over-expression vector), were amplified from cDNA of *TM-1* and integrated into the linear pCAMBIA1300 vector containing a superfolder green fluorescent protein (sGFP) gene (digested by KpnI and SalI), thus constructing the CaMV35S driving and GENE-sGFP-fused expression vector. *Gh_A05G3028* and *Gh_D10G2310* fused GFP genes were transiently expressed in epidermal cells of tobacco via *Agrobacterium* with expressed nuclear localization sequence fused to a reporter gene mCherry as a nuclear marker. The GFP and red fluorescent protein fluorescence in epidermal cells of tobacco was detected and photographed using a fluorescence microscope Imager A2 (Zeiss, oberkochen, Germany). The primers Gh_A05G3028-CDS-U/L and Gh_D10G2310-CDS-U/L are listed in Appendix A.

### 4.8. Gene Ontology Enrichment and Kyoto Encyclopedia of Genes and Genomes (KEGG) Analysis

To better understand the regulatory mechanisms of RCC1 family genes and salt tolerance, ccNET (http://structuralbiology.cau.edu.cn/gossypium/) was used for co-expression analysis. Meanwhile, the cotton FGD (https://cottonfgd.org/analyze/) website was used for gene ontology (GO) enrichment and Kyoto Encyclopedia of Genes and Genomes (KEGG) analysis.

### 4.9. VIGS (Virus-Induced Gene Silencing) of the Gh_A05G3028 and Gh_D10G2310 in Cotton

The TRV (tobacco rattle virus) constructs *pTRV-RNA1* and *pTRV-RNA2* were prepared according to the methods of Fradin [49]. The fragments targeting to the *Gh_A05G3028*, *Gh_D10G2310* gene and containing KpnI and EcoRI recognition sites were amplified using cDNA generating from RNA of *TM-1* as a template and integrated into pTRV2, thus constructing the *TRV2: Gh_A05G3028* and *TRV2: Gh_D10G2310* vectors. The *pTRV2*-fused cDNA fragment of *GhCLA* (cloroplastos alterados 1) was used as a positive control to monitor the silencing efficiency. Then the recombinant plasmids of *TRV: Gh_A05G3028*, *TRV: Gh_D10G2310* vectors, and *TRV: GhCLA* were transferred into Agrobacterium tumefaciens (GV3101). The seeds of *TM-1* were germinated in soil under the conditions: 12,000 Lx light 12 h at 23 °C/dark 12 h at 23 °C, 80% humidity for 7 days. The cotyledons of cotton were injected with an infective solution containing Agrobacterium tumefaciens of *pTVR1* and *TRV2: GhCLA* or *pTRV1* and *TRV2: target gene*. Cotton infected by *Agrobacterium tumefaciens* was cultured under the conditions above until the white-striped leaf or albino phenotype appeared in the positive control. The negative control (*pTRV1* coupled with *pTRV2*) and experimental group (*pTRV1* coupled with *TRV: Gh_A05G3028* or *TRV: Gh_D10G2310*). Some of the plants were watered with 1/2 MS nutrient solution as the control and the others were watered with 200 mM salt solution regularly every 3 days until the phenotypes appeared. The primers for the VIGS assay are listed in Appendix A.

### 4.10. Determination of Chlorophyll Content

Fifty milligrams leaves of Arabidopsis thaliana or cotton were weighed and added into a 10 mL centrifugal tube, sealed with 2.5 mL 95% ethanol, and extracted at room temperature for 36 h under dark conditions. The extract solution was diluted two times, and then colorimetric analysis was carried out by the SYNERGY H1 microplate reader (BioTeK, Winooski, VT, USA) at 665 and 649 nm, respectively. The concentration of chlorophyll a and b (mg·L^−1^) was calculated as the following formula, and the content of each pigment (mg·g^−1^) was calculated according to the calculated concentration:
Ca = 13.95 × OD_665_−6.88 × OD_649_(1)
Cb = 24.96 × OD_649_−7.32 × OD_665_(2)

The content of pigments in leaves (mg/g) = concentration of pigments (m g/L) × total volume of extracts (ml) × dilution multiple/sample quality (g).

## 5. Conclusions

In Conclusion, 56 *RCC1* family genes in *G. hirsutum* were identified and comprehensively analyzed in terms of gene structure, chromosome distribution, conserved domains, phylogenetic relationship and expression patterns in different tissues. Besides, the expression patterns of 56 *RCC1* family genes in *G. hirsutum* under various hormones and abiotic stresses were further analyzed and revealed that *Gh_A05G3028* and *Gh_D10G2310* were significantly induced by salt. The phenotypic analysis of their homologous mutants in *Arabidopsis* and VIGS lines in *G. hirsutum* under salt treatment further proved their crucial role in salt tolerance of cotton. Additionally, the co-expression network analysis of the two genes revealed their possible downstream genes in the regulation of salt tolerance in cotton. The results above provide valuable reference data for further study of *RCC1* genes in cotton.

## Figures and Tables

**Figure 1 ijms-20-01727-f001:**
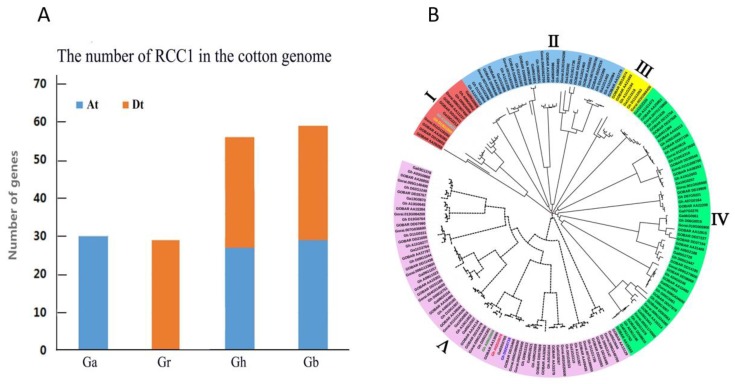
Number and phylogenetic analysis of the *RCC1* gene family in four cotton species, *G. arboreum* (Ga), Gr = *G. raimondii*, Gh = *G. hirsutum*, Gb = *G. barbadense*, At = A subgroup, Dt = D subgroup. (**A**) Comparison of the number of *RCC1* genes in four cotton varieties. (**B**) Non-root phylogenetic tree of the *RCC1* protein sequence of *G. arboreum*, *G. raimondii*, *G. hirsutum*, and *G. barbadense.* The full-length protein sequences from several plant species were aligned by CLUSTAL Omega and the phylogenetic tree was built using the adjacency (neighbor-joining) method in the iTOLv4 website. The bootstrap value was 1000 replicates.

**Figure 2 ijms-20-01727-f002:**
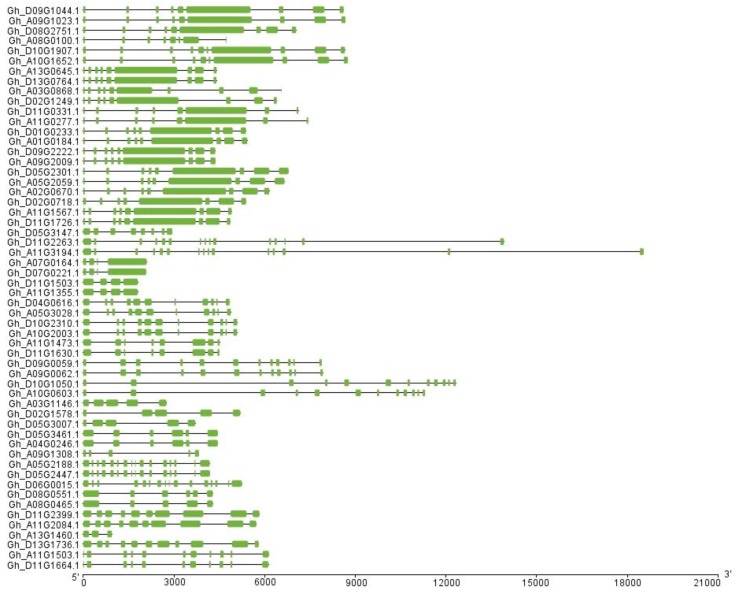
The exon/intron organization of RCC1 genes of *G. hirsutum*. (**Left**) *GhRCC1* family gene names. Green bars = exons, black lines = introns. Each bar section = 3 kb.

**Figure 3 ijms-20-01727-f003:**
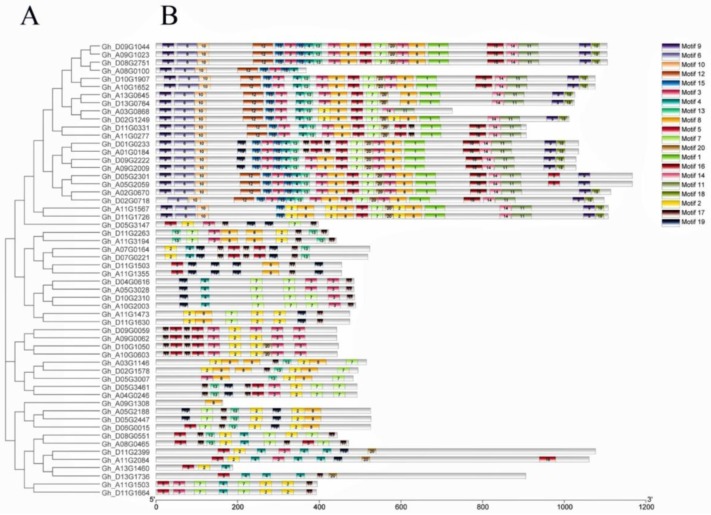
Phylogenetic and conservative motifs analysis of the RCC1 gene family in *G. hirsutum*. (**A**) The phylogenetic tree of all *RCC1* genes in *G. hirsutum* was constructed using the adjacency method and the bootstrap test was performed with 1000 iterations. (**B**) The conserved protein motifs in the *RCC1* family were identified using the MEME program. Each motif is represented by a different color.

**Figure 4 ijms-20-01727-f004:**
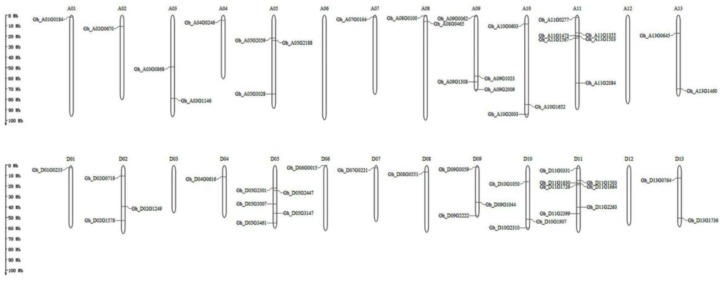
Chromosomal distribution of RCC1 genes in *G. hirsutum*. The chromosome number is located directly above each vertical bar. Two genes were found on unassembled scaffolds (*Gh_D08G2751* and *Gh_A11G3194*). The scale is in Mb.

**Figure 5 ijms-20-01727-f005:**
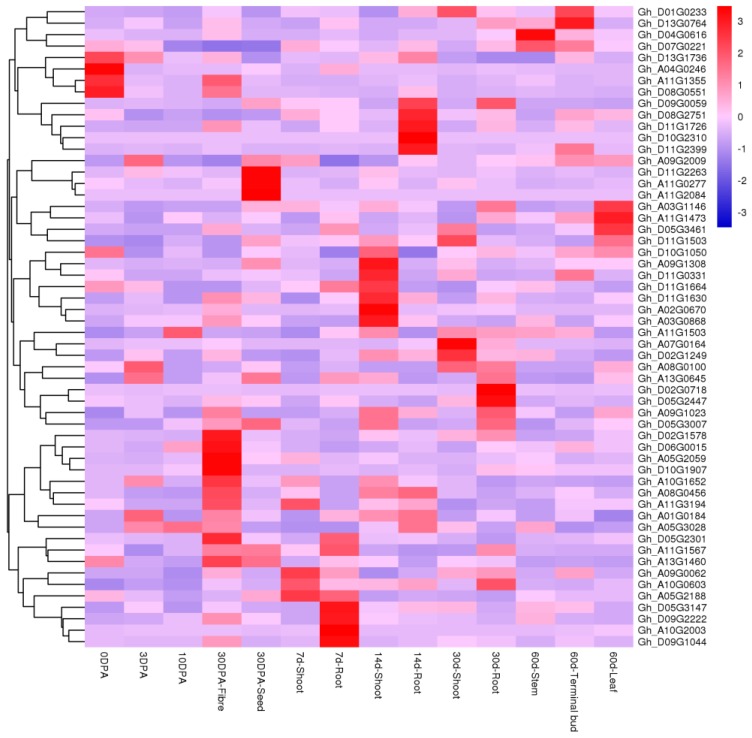
Heat map showing hierarchical clustering of expression levels in various tissues of upland cotton. (**Top right**) Cluster analysis of gene expression levels at different color scales. d = Day, 0 DPA = 0 day post-anthesis ovules, 3 DPA = day post-anthesis ovules, 10 DPA = 10 day post-anthesis ovules, 30 DPA-fiber = 30 day post-anthesis fibers, 30 DPA-seed = 30 day post-anthesis seed, shoot = above ground, root = underground. The relevant gene expression data were normalized.

**Figure 6 ijms-20-01727-f006:**
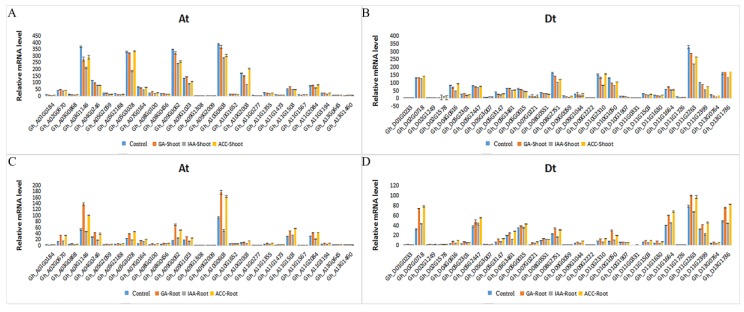
Expression of *GhRCC1* genes in seedlings following hormone treatment. (**A**) Expression profiles of *GhRCC1* At family genes and (**B**) Dt family genes in shoots treated with 50 mM IAA (indole-3-acetic acid), 5 mM GA (gibberellic acid), or 50 mM ACC (1-aminocyclopropane-1-carboxylic acid) for 3 h, respectively. (**C**) Expression profiles of *GhRCC1* At family genes and (**D**) Dt family genes in roots treated with 50 mM IAA, 5 mM GA, or 50 mM ACC for 3 h, respectively. Error bars represent standard deviations from three biological replicates.

**Figure 7 ijms-20-01727-f007:**
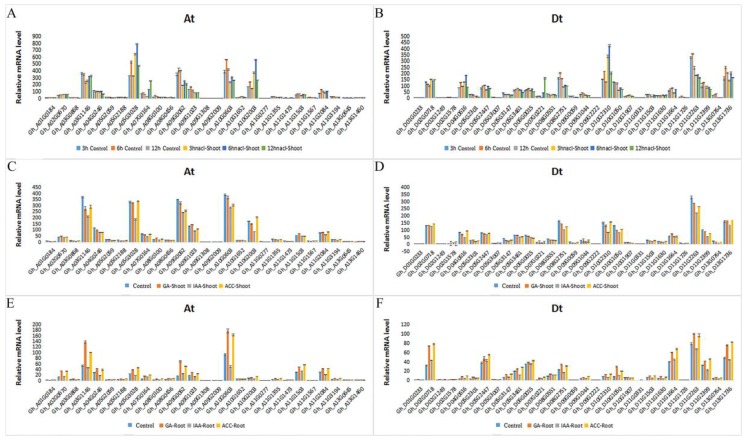
Expression profile analysis of *GhRCC1* genes’ response to salt and drought stress. Gene expression analysis of the *GhRCC1* family in cotton seedlings grown for 10 days. (**A**) Expression pattern of *GhRCC1* At subfamily gene after treatment of cotton seedlings for 3, 6, and 12 h with 200 mM NaCl. (**B**) Expression pattern of the *GhRCC1* Dt subfamily gene after treatment of cotton seedlings for 3 h with 200 mM NaCl. (**C**) Expression profiling of the *GhRCC1* At subfamily genes and (**D**) Dt subfamily genes in shoots treated with 17% PEG6000. (**E**) Expression profiling of the *GhRCC1* At subfamily genes and (**F**) Dt subfamily genes in roots treated with 17% PEG6000. Error bars represent standard deviations from three biological replicates.

**Figure 8 ijms-20-01727-f008:**
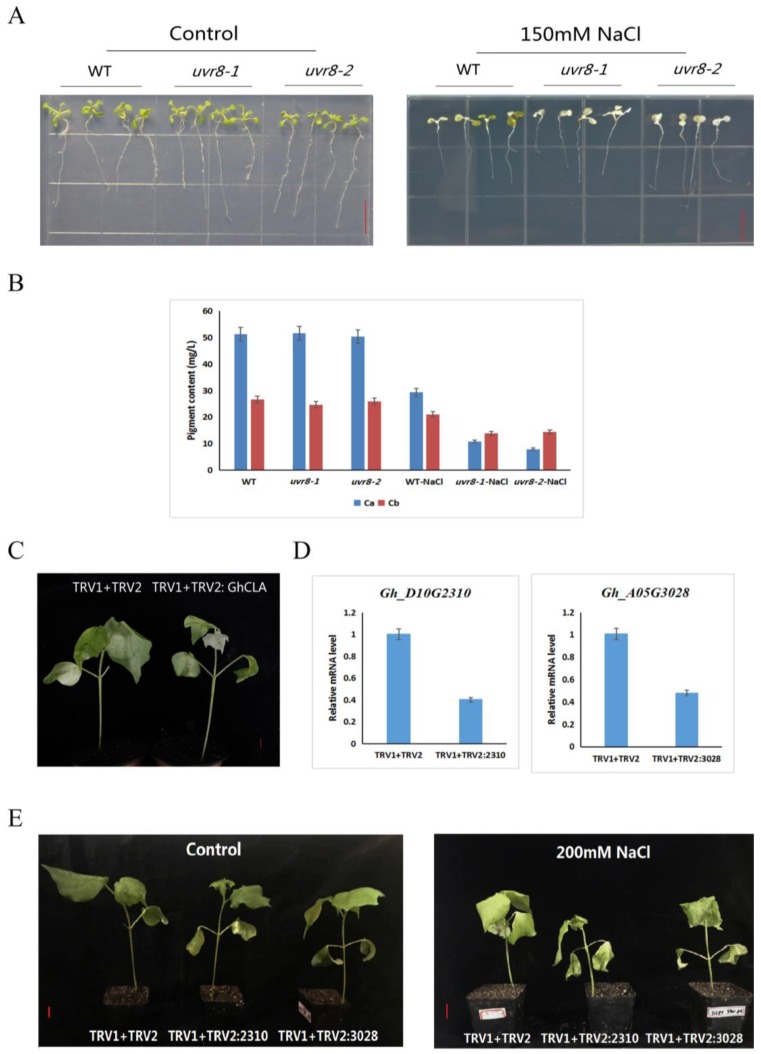
Phenotype of *UVR8* mutants and *Gh_A05G3028* and *Gh_D10G2310* VIGS (virus induced gene silencing) plants under salt treatment. (**A**) Phenotypes of WT (wild-type) and *UVR8* mutants of 7-day-old seedlings grown on 1/2 Murashige and Skoog (MS) media (control) and 1/2 MS media containing 150 um NaCl. WT: Col-0, *uvr8-1* (SALK_072594C)/*uvr8-2* (SALK_072594C): T-DNA insertional mutants for *UVR8* under the background of Col-0. (**B**) Determination of chlorophyll content in the wild-type and *UVR8* mutants of Arabidopsis thaliana. Ca: chlorophyll a, Cb: chlorophyll b. Error bars represent the standard deviation of the three biological replicates. (**C**) Albino phenotypes in GhCLA1 VIGS plants. TRV1 + TRV2: negative control, TRV1 + TRV2: GhCLA: *GhCLA1* VIGS plants. (**D**) Fluorescence quantitative PCR analysis of expression of *Gh_A05G3028* and Gh_*D10G2310* in control and VIGS plants. TRV1 + TRV2: negative control, TRV1 + TRV2: 3028 and 2310: *Gh_A05G3028* and *Gh_D10G2310* VIGS plants. Error bars represent the standard deviation of the three biological replicates. (**E**) Phenotype of *Gh_A05G3028* and *Gh_D10G2310* VIGS seedlings under control and salt stress. All red bars are 1 cm.

**Figure 9 ijms-20-01727-f009:**
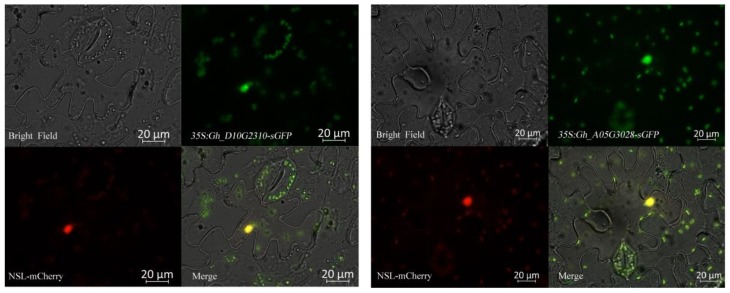
Subcellular localization of two selected GhRCC1 proteins. GhRCC1 and GFP (green fluorescent protein)-fusion proteins were transiently expressed in epidermal cells of tobacco driven by the CaMV35S promoter. Nuclear localization sequence-fused reporter gene mCherry was used as a nucleus-located marker. Bar = 20 μm.

**Figure 10 ijms-20-01727-f010:**
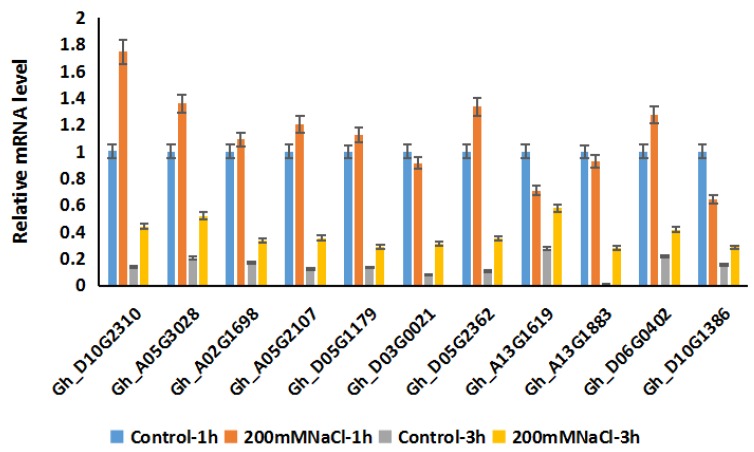
Expression profile analysis of *Gh_D10G2310*, *Gh_A05G3028,* and co-expressed genes under salt stress. The expression levels of *Gh_D10G2310*, *Gh_A05G3028,* and co-expressed genes related to salt resistance under control and salt stress at 1 and 3 h. Control-1 h: 1 h after watering with 1/2 MS liquid medium, NaCl-1 h: 1h after watering with 1/2 MS liquid medium containing 200 μm NaCl, Control-3 h: 3 h after watering with 1/2 MS liquid medium, NaCl-3 h: 3 h after watering with 1/2 MS liquid medium containing 200 μm NaCl. Error bars represent the standard deviation of the three biological replicates.

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
