# Peer review of "Identification and Expression Profiling of the *Regulator of Chromosome Condensation 1 (RCC1)* Gene Family in *Gossypium Hirsutum* L. under Abiotic Stress and Hormone Treatments"

_ijms, 2019, doi:10.3390/ijms20071727_

Round 1

Reviewer 1 Report

I think this is impressive work presented as much as possible to be error free.

Line 32:  "...arable land area available to these crops decreases year on year." better read "...arable land area available to these crops decreases each year."

Line 33: "...Due to the lack of vacant farm land for..." Could better read "...Due to insufficient farm land for..."

Line 40: "...damage to membrane systems, and, ultimately, apoptosis..." The comma after and maybe better removed.  "...damage to membrane systems, and ultimately, apoptosis..."

Author Response

Dear reviewer,

Re: [IJMS] Manuscript ID: ijms-459416 - Major Revisions

Thank you for your valuable suggestion. Here are my responses to the your comments.

Comment 1: Line 32:  "...arable land area available to these crops decreases year on year." better read "...arable land area available to these crops decreases each year."

Response:We have revised it in accordance with the suggestion. The revised position is in line 32.

Comments 2: Line 33: "...Due to the lack of vacant farm land for..." Could better read "...Due to insufficient farm land for..."

Response:We have revised it in accordance with the suggestion. The revised position is in line 33.

Comments 3: Line 40: "...damage to membrane systems, and, ultimately, apoptosis..." The comma after and maybe better removed.  "...damage to membrane systems, and ultimately, apoptosis..."
Response:We have revised it in accordance with the suggestion. The revised position is in line 40.

We have revised the manuscript accordingly and the additional parts are marked in red font and deleted parts are marked with blue deletion lines. I hope this will make it more acceptable for publication.

Yours sincerely,

Chendong Sun

Reviewer 2 Report

Major issues that must be addressed:This work is adequately written and the results are interesting. However, there are inconsistencies in the document that must be addressed (described below)

-Inconsistencies between material and methods (M&M) and the results:

Figure 7 shows treatments with PEG6000 but these experiments are not described in M&M. Only salt treatment is described (lines 104-105). Furthermore, Figure 7 legend says that seedlings were grown 10 days but in M&M are 7 days seedlings. The same figure legend describes only 3 hours salt treatment in fig7.A and fig7.B but you can see treatments at 3h, 6h and 12h in the figures. For more inconsistency, M&M describes the salt treatment at 3h and 6h but not 12h and nothing is said in M&M about 17% PEG 6000 treatments.

Figure 10 legend also describes 10 day-old seedlings and in the graph the salt treatment is 1h_NaCl. However, in M&M the salt treatment described is 3h and 6 hours for 7 day-old seedlings.

-Inconsistencies in the results:

In line 321 it is stated that mutants uvr8-1 and uvr8-2 exhibited salt-sensitive phenotypes in comparison with the wild type (fig.8). However, only uvr8-1 and not uvr8-2 it is shown in figure 8. On top of that, fig.8 legend highlight the bars length ("all bars are 1cm"). Are you referring to the red bars? Why the length is relevant? The roots in fig B look the same length. Authors should specify the parameters to measure the salt sensitiveness of the plants. If the parameter is the greenness of the plant or the size of the leaves or the number of secondary roots, some quantification and statistics would help to support your conclusions.

In lines 345-348 you talk about 20 genes in fig.S2 but in figure 2 in supplementary data there are only 7 genes. On top of that, the text implies that you select 2 genes out of those 20 (Gh_D10G1386 and Gh_D06G0402) for qPCR in your samples but you don't say why. Furthermore, those 2 genes selected show the lowest expression level among the 7 genes plotted in fig.S2 and any differences can be appreciated. Consistently with these observation, there are not big differences of expression between control and salt treatment in fig.10. Statistics should be added to fig. 10 (the same you did in fig.S1C) in order to support your conclusions.

-Inconsistencies between the results and the discussion/conclusions:

In lines 399-400 you state that most of genes increased in their expression in the underground part (I guess it means in the root) after GA treatment but in results the effect of ethylene is also detected (both in the text and in fig 6)

Lines419-420. The figure 8 should be improved to support that conclusion. As a minimum, the second mutant must be included in fig.8 to be able to use the term "mutants" in plural.

Lines 423-424. Co-expression KEGG analysis and GO term analysis are used in this work as a important source of knowledge in this work. It is described in M&M and it is mentioned in both results and discussion. However, data are not shown, nor the reason why those 20 genes were supposed to be shown in fig.S2 were selected. I suggest to include co-expression analysis at least in supplementary data.

 2. Accuracy improvement so the experiments can be replicated by other researchers:

From line 145 to 150 components are described in uL(microliters), but the concentration of RNA, DNAase, cDNA and primers is unknown. I suggest to provide the amount of cDNA and primer pair and also the number of units of DNAse (or at least the brand and reference number). 

 3. Minor text editing suggestions

Line 159- A full stop is needed between "method" and "Primers". 

Line 187- "And" should be eliminated after a full stop.

Line 207- I suggest to replace the "least" by the "lowest".

Line 211- "(Fig.2) should be at the end of the sentence.

Line 330- "in epidermal cells of tobacco" could be added after "assays".

Line 346- "the above" could be added before "20 genes".

Line 3371- Remove space before "." (maybe there are also 2 spaces in line 369 before "(Abe et al."

Lines 405-413 That paragraph could be placed in introduction, rather than in discussion.

Lines 417-419 "although the amino...". This is reiterative (it is already explained in results)

4. Other suggestions that might improve the study (not required)

Apparently motif number3 is present in all RCC1 gene family (Fig.3). I missed some discussion about the significance of the conserved motifs results in the discussion.

Figure 5 is a little confusing because all the tissues are together. You could ask yourselves if the cluster analysis would give you more information if comparing samples in 3 blocks (0DPA-30DPA, 7d-shoot-30Droot and 60dStem-60dLeaf).

Lines 427-430. Here there is a strong conclusion not fully supported by the data, specially taking in account that you used uvr8 mutants instead of tcf1 mutants. To measure the levels of the cotton homologous for H3K4me3 and H3K27me3 in your samples where Gh_A05G3028 and GhD10G2310 are overexpressed could improve this work.

Author Response

Dear reviewer,

Re: [IJMS] Manuscript ID: ijms-459416 - Major Revisions

Thank you for your valuable suggestion. Here are my responses to your comments.

Comments 1:Figure 7 shows treatments with PEG6000 but these experiments are not described in M&M.

Response: First of all, we have to apologize for our negligence in writing. Indeed, there is a method of PEG treatment in the pre-revision version, but we accidentally deleted it in the revision process. And we have added the PEG treatment method into “Plant materials and treatment”. The revised position is in line 107.

Comment 2: Furthermore, Figure 7 legend says that seedlings were grown 10 days but in M&M are 7 days seedlings.
Response: In fact, we used 10-day seedlings for phytohormone, PEG and salt treatment. Thus, we have revised it in “Plant materials and treatment”. The revised position is in line 102.

Comment 3: The same figure legend describes only 3 hours salt treatment in fig7. A and fig7.B but you can see treatments at 3h, 6h and 12h in the figures.

Response: In fact, we analyzed the gene expression patterns at 3h, 6h and 12h under salt treatment, and the results were shown in Figure 7. However, we omitted 6h and 12h when described in the figure legend. We have added 6h and 12h into the figure legend.

Comment 4: For more inconsistency, M&M describes the salt treatment at 3h and 6h but not 12h and nothing is said in M&M about 17% PEG 6000 treatments.

Response: We have revised it in “Plant materials and treatment”. The revised position is from line 106 to line 107.

Comment 5: Figure 10 legend also describes 10 day-old seedlings and in the graph the salt treatment is 1h_NaCl. However, in M&M the salt treatment described is 3h and 6 hours for 7 day-old seedlings.

Response: We have revised it in “Plant materials and treatment”. The revised position is in line 102 and line 107.

Comment 6: In line 321 it is stated that mutants uvr8-1 and uvr8-2 exhibited salt-sensitive phenotypes in comparison with the wild type (fig.8). However, only uvr8-1 and not uvr8-2 it is shown in figure 8.

Response: In the last experiment, although we observed that both mutants exhibited salt-sensitive, we did not show the phenotype of uvr8-2 because the medium for uvr8-2 was infected by bacteria. We have re-performed the experiment and the results shown it in Figure 8.

Comment 7: On top of that, fig.8 legend highlight the bars length ("all bars are 25px"). Are you referring to the red bars? Why the length is relevant? The roots in fig B look the same length.

Response: The length of the bar is correct. Intracellular oxygen species (ROS), produced by high concentration of salt in the environment, will inhibit mitotic activity of root apical meristem (RAM). Plants under salt treatment are shorter than those under normal culture conditions. Thus, the bars in figure8 A and B are equal but root length is different.

Comment 8: Authors should specify the parameters to measure the salt sensitiveness of the plants. If the parameter is the greenness of the plant or the size of the leaves or the number of secondary roots, some quantification and statistics would help to support your conclusions.

Response: Yes, we have performed determination of chlorophyll content assay to explain the salt-sensitive phenotype of the mutants in figure8.

Comment 9: In lines 345-348 you talk about 20 genes in fig.S2 but in figure 2 in supplementary data there are only 7 genes. On top of that, the text implies that you select 2 genes out of those 20 (Gh_D10G1386 and Gh_D06G0402) for qPCR in your samples but you don't say why. Furthermore, those 2 genes selected show the lowest expression level among the 7 genes plotted in fig.S2 and any differences can be appreciated. Consistently with these observation, there are not big differences of expression between control and salt treatment in fig.10. Statistics should be added to fig. 10 (the same you did in fig.S1C) in order to support your conclusions.

Response: some genes are involved in multiple signaling pathways related to salt resistance of plants, thus leading to repeated calculations of genes. Actually, we filtered it out in the database and 16 gene is closely related to the expression of Gh_A05G3028 and Gh_D10G2310, and plays an important role in salt tolerance of cotton. The FPKM from the database of ccNET reveals that 10 genes among them were up-regulated under salt stress in comparison to control. Nevertheless, 1 gene among the 10 genes is hardly expressed under control and salt treatment. (FPKM<10). To further confirm the expression profile of these 9 genes under salt stress, we conducted RT-qPCR. However, the previous data showed that only 2 genes were slight up-regulated under salt stress, probably resulting from the water culture method. As is known, available ion concentration in water is higher than that in soil, so cotton will be exposed to more sodium ions in hydroponic environment, leading to more ROS production in internal environment and finally resulting in the fracture of nucleic acid (DNA and RNA). Therefore, the down-regulation of gene expression probably due to the incompleteness of nucleic acid rather than salt-induced down-regulation. And we replaced hydroponics with soil culture and further analyzed expression pattern of the 9 genes and Gh_A05G3028, Gh_D10G2310 under control and salt treatment at 1h and 3h, the results showed in figure10 conformed that all genes exhibited a salt-induced expression parttern.

Comment 10: In lines 399-400 you state that most of genes increased in their expression in the underground part (I guess it means in the root) after GA treatment but in results the effect of ethylene is also detected (both in the text and in fig 6)

Response: First of all, the underground part indeed means the root. And your comments are very accurate, not only under GA treatment but under ethylene treatment, most of genes increased in their expression in the underground part, thus we have revised and added the ethylene treatment into the discussion. The revised position is in line 481-482.

Comment 11: Lines419-420. The figure 8 should be improved to support that conclusion. As a minimum, the second mutant must be included in fig.8 to be able to use the term "mutants" in plural.

Response: We have added the phenotype of second mutant and determined the chlorophyll content under control and salt treatment. Furthermore, two target genes were silenced using VIGS (virus induced gene silencing) system. As the results, the gene silenced cotton exhibited a salt sensitivity comparing with the negative control under salt treatment, further confirmed our conclusion. All results were shown in figure8.

Comment 12: Lines 423-424. Co-expression KEGG analysis and GO term analysis are used in this work as a important source of knowledge in this work. It is described in M&M and it is mentioned in both results and discussion. However, data are not shown, nor the reason why those 20 genes were supposed to be shown in fig.S2 were selected. I suggest to include co-expression analysis at least in supplementary data.

Response: We have revised it according to your suggestion. The FPKM of the 16 genes related to salt resistance under control and salt treatment from ccNET was listed in supplementary data.

Comment 13: From line 145 to 150 components are described in uL(microliters), but the concentration of RNA, DNAase, cDNA and primers is unknown. I suggest to provide the amount of cDNA and primer pair and also the number of units of DNAse (or at least the brand and reference number).

Response: We have revised it according to your suggestion. The revised position is in line 156-160.

Comment 14: Line 159- A full stop is needed between "method" and "Primers".

Response: We have revised it according to your suggestion. The revised position is in line 172.

Comment 15: Line 187- "And" should be eliminated after a full stop.

Response: We have revised it according to your suggestion. The revised position is in line 229.

Comment 16: Line 207- I suggest to replace the "least" by the "lowest".

Response: We have revised it according to your suggestion. The revised position is in line 250.

Comment 17: Line 211- "(Fig.2) should be at the end of the sentence.

Response: We have revised it according to your suggestion. The revised position is in line 251.

Comment 18: Line 330- "in epidermal cells of tobacco" could be added after "assays".

Response: We have revised it according to your suggestion. The revised position is in line 405.

Comment 19: Line 346- "the above" could be added before "20 genes".

Response: We have revised it according to your suggestion. The revised position is in line 420.

Comment 20: Line 3371- Remove space before "." (maybe there are also 2 spaces in line 369 before "(Abe et al."

Response: We have revised it according to your suggestion. The revised position is in line 451-452.

Comment 21: Lines 405-413 That paragraph could be placed in introduction, rather than in discussion.

Response: Your suggestion is reasonable, and we have considered it for a long time, but finally, we did not revise it for two reasons: Firstly, deleting this section will result in incoherence in the discussion. At the same time, the following discussion of the salt resistance mechanism of these two RCC1 family genes is involved in this part of content. Secondly, this paragraph and the introduction have some repetition. Thus, adding this paragraph to the introduction will cause redundancy. We sincerely hope you can understand us.

Comment 22: Lines 417-419 "although the amino...". This is reiterative (it is already explained in results)

Response: Yes, you are right. We have deleted it according to your suggestion.

We have revised the manuscript accordingly and the additional parts are marked in red font and deleted parts are marked with blue deletion lines. I hope this will make it more acceptable for publication.

Yours sincerely,

Chendong Sun

Round 2

Reviewer 2 Report

You made a great effort to improve the paper.

I am still missing some statistics on the histograms added in Fig.8 and the new fig 10 but I think the new figures are clear enough to make your point.